# Keratinocytes contribute to normal cold and heat sensation

**Katelyn E Sadler[†], Francie Moehring[†], Cheryl L Stucky***

Department of Cell Biology, Neurobiology, and Anatomy, Medical College of Wisconsin, Milwaukee, United States

**Abstract** Keratinocytes are the most abundant cell type in the epidermis, the most superficial layer of skin. Historically, epidermal-innervating sensory neurons were thought to be the exclusive detectors and transmitters of environmental stimuli. However, recent work from our lab (Moehring et al., 2018) and others (Baumbauer et al., 2015) has demonstrated that keratinocytes are also critical for normal mechanotransduction and mechanically-evoked behavioral responses in mice. Here, we asked whether keratinocyte activity is also required for normal cold and heat sensation. Using calcium imaging, we determined that keratinocyte cold activity is conserved across mammalian species and requires the release of intracellular calcium through one or more unknown cold-sensitive proteins. Both epidermal cell optogenetic inhibition and interruption of ATP-P2X4 signaling reduced reflexive behavioral responses to cold and heat stimuli. Based on these data and our previous findings, keratinocyte purinergic signaling is a modality-conserved amplification system that is required for normal somatosensation in vivo.

**\*For correspondence:**
cstucky@mcw.edu

[†]These authors contributed equally to this work

**Competing interests:** The authors declare that no competing interests exist.

## Introduction

Skin is the largest sensory organ. As the most superficial layer of the skin, the epidermis is the primary tissue that interfaces with environmental stimuli. Specialized structures within the epidermis including Meissner corpuscles and Merkel cells couple with Aβ low-threshold sensory afferents to encode unique qualities of mechanical force to the central nervous system. These specialized epidermal cells have received significantly more attention in somatosensory research than keratinocytes, the so-called 'barrier cells' that constitute 95% of the epidermis. Recently however, we and the team of Baumbauer, DeBerry, and Adelman et al. reported that keratinocytes are required for normal innocuous and noxious mechanosensation; optogenetic inhibition of keratin14-expressing cells decreases mechanically-induced firing in Aβ, Aδ, and C fiber afferents (*Baumbauer et al., 2015*) and behavioral responses to hindpaw mechanical stimulation in naïve mice (*Moehring et al., 2018*). However, it is still unclear if keratinocyte activity is also required for normal cold and heat sensation.

Very little is known about the molecular basis or functional importance of keratinocytes in cold transduction and sensation, despite regular epidermal exposure to decreasing environmental temperatures. Primary keratinocytes from both mouse and human exhibit cold-induced calcium transients (*Bidaux et al., 2015*; *Tsutsumi et al., 2010*), but a direct comparison of these responses has not yet been performed. In this report, we measured cold-induced calcium transients in keratinocytes isolated from mouse, rat, hibernating 13-lined ground squirrel, and human tissue donors to determine whether keratinocyte cold sensitivity is conserved between mammalian species with different thermostatic regulation mechanisms, and to assess the translatability of findings obtained in these model organisms. We also utilized a variety of chemical tools and transgenic animal lines to characterize the molecular cold transducer in mouse keratinocytes. Finally, we employed global inhibitory optogenetic approaches to assess if keratinocyte activity is required for in vivo cold and heat sensation; optogenetic inhibition of keratin14-expressing cells increased withdrawal latencies to both cold and heat stimuli. Similar to our previous mechanosensation studies (*Moehring et al.,*

*2018*), we also observed decreased cold and heat sensitivity in wildtype mice after intraplantar injections of apyrase, an enzyme that catalyzes ATP hydrolysis, and in sensory-neuron specific P2X4 receptor knockout mice (P2X4$^{cKO}$). Collectively, these experiments are the first to demonstrate an in vivo requirement of keratinocyte activity for normal cold and heat sensation. Based on these and our previous studies (*Moehring et al., 2018*), we conclude that purinergic keratinocyte-to-sensory neuron signaling is a modality-conserved mechanism that serves as a critical amplifier for mechanical, cold, and heat sensation.

## Results and discussion

### Mammalian keratinocyte cold responses differ between species

Temperature detection is critical for all species' survival. In conventional theories of somatosensation, primary sensory afferents are the initial detectors of environmental stimuli; increases and decreases in temperature are primarily detected by subclasses of polymodal C and Aδ fibers. This theory, which differs from the requirement of non-neuronal cells in other sensory transduction circuits (e.g. taste cells in gustation, hair cells in auditory and vestibular senses, photoreceptors in vision), is not supported by mechanotransduction studies from our group and others; keratinocytes are not only capable of responding to mechanical force, but are required for normal mechanosensation in vivo (*Baumbauer et al., 2015*; *Moehring et al., 2018*). To determine if keratinocytes are required for cold sensation, we first examined the inherent cold sensitivity of these cells from multiple rodent model organisms and humans. Calcium transients were compared in keratinocytes isolated from mouse, rat, and hibernating 13-lined ground squirrel glabrous hindpaw skin and human hairy skin while external buffer temperature was decreased (~24°C to 12°C over 1 min; *Figure 1A*). All cold-responsive keratinocytes exhibited sustained calcium transients that began during the initial dynamic phase of the cold ramp and ended during the static cold hold phase. Cytosolic calcium levels began to decrease during the static phase of the temperature ramp, and a second increase in cytosolic calcium was not observed during re-warming. This could be because the keratinocytes did not establish a new thermostatic set-point during the 2 min static cold hold. If keratinocytes quickly reset their set-point at the temperature nadir (~12°C), we would have expected warming responses to occur during the second dynamic phase of the ramp. However, we observed such a warming response in <1% of mouse keratinocytes.

Keratinocyte cold responses varied between the species tested. Over 92% of mouse keratinocytes responded to decreasing temperatures (*Figure 1B*). On average, the magnitude of the peak cold response was a 56% increase from baseline (*Figure 1C*), and cold-induced transients were initiated when the buffer temperature dropped below 20.9°C (a 3°C drop from the starting temperature; *Figure 1D*). Rat keratinocytes were significantly less responsive to cold. Only 47% of rat keratinocytes exhibited cold-induced calcium transients, the peak magnitudes and temperature thresholds of which were lower than mouse (*Figure 1*). The biological relevance of this species differences is unclear. *Mus musculus* and *Rattus novegicus* are members of the *Muridae* family that inhabit similar climates (*Feng and Himsworth, 2014*; *Latham and Mason, 2004*) and maintain similar core and skin temperatures. Both species exhibit similar reflexive and thermotactic behaviors when exposed to cold stimuli or environments, and ex vivo skin-nerve recordings have identified spontaneously active, low-threshold cool fibers and quiescent, high-threshold cold fibers (*Leem et al., 1993*; *Koltzenburg et al., 1997*; *Paricio-Montesinos et al., 2020*) in both species. To our knowledge, direct comparisons of behavioral and physiological cold responses between mouse and rat do not exist, but if completed, may reveal nuanced species differences that could be keratinocyte-mediated.

Cold responses were also assessed in keratinocytes isolated from 13-lined ground squirrels (*Ictidomys tridecemlineatus*) in hibernation. Over 96% of squirrel keratinocytes responded to decreasing temperatures (*Figure 1B*). Cold-induced calcium transients in squirrel keratinocytes were larger than those observed in mouse or rat and were initiated at a threshold of ~19°C (*Figure 1*). 13-lined ground squirrels are obligate hibernators indigenous to midwestern North America. In laboratory behavioral studies, these rodents are less sensitive to innocuous and noxious cold than mice as a

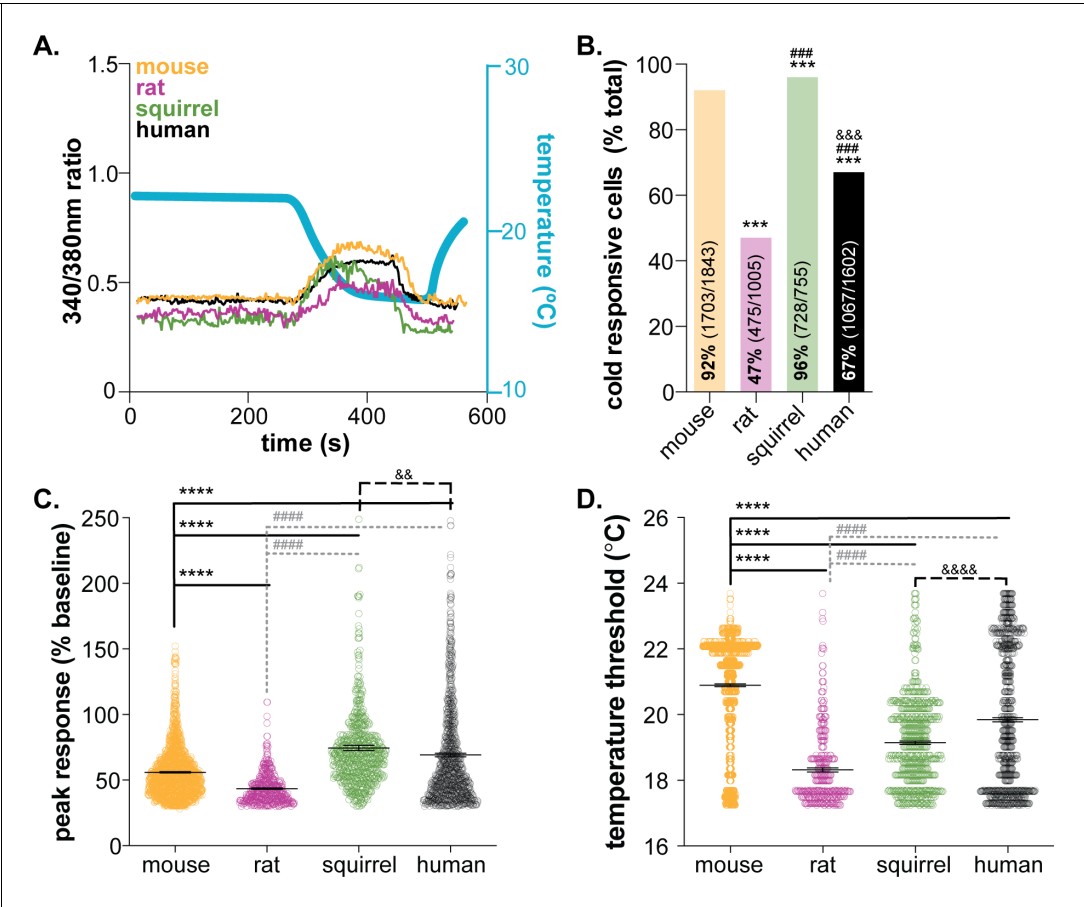

**Figure 1.** Mammalian keratinocytes respond to temperature decreases. In vitro calcium imaging was performed on primary cultured keratinocytes from the glabrous hindpaw skin of C57BL/6 mice, Sprague Dawley rats, and 13-lined ground squirrels and human breast skin. (**A**) Representative keratinocytes from all species exhibited a calcium transient upon extracellular buffer cooling (~23°C to 12°C). (**B**) The greatest proportion of cold-responsive cells was observed in hibernating 13-lined ground squirrel samples. The vast majority of mouse keratinocytes also responded to decreasing temperatures; fewer rat and human keratinocytes responded to cold (Chi-square p<0.0001; Fisher's Exact tests: ***p<0.001 vs. mouse; ### p<0.001 vs. rat; &&& p<0.001 vs. squirrel; n = 28 mice, four rats, four squirrels, five humans). (**C**) The peak calcium response to decreasing temperatures varied between species; hibernating 13-lined ground squirrel keratinocytes exhibited the largest calcium transients (75% increase over baseline) and Sprague Dawley rats exhibited the smallest (44% increase over baseline; mouse: 56% increase; human: 69% increase; 1-way ANOVA p<0.0001; Bonferroni's multiple comparisons: ****p<0.0001 vs. mouse; #### p<0.0001 vs. rat, && p=0.0084 vs. squirrel). (**D**) The temperature at which keratinocytes responded to cold (>30% increase in Δ340/380) differed between species. Mean temperature thresholds (°C) for mouse: 20.9, rat: 18.3, squirrel: 19.1, human: 19.8 (Kruskal-Wallis test p<0.0001; Dunn's multiple comparisons: ****p<0.0001 vs. mouse, #### p<0.0001 vs. rat, &&&& p<0.0001 vs. squirrel).
The online version of this article includes the following figure supplement(s) for figure 1:

**Figure supplement 1.** Variability in human keratinocyte responses to temperature decreases.

result of amino acid substitutions in the TRPM8 channel and decreased cold-sensitivity of dorsal root ganglia neurons (*Matos-Cruz et al., 2017*). Suppressed cold sensation is proposed to be metabolically advantageous for species that spend extended periods of time in hibernation (*Matos-Cruz et al., 2017*). We were therefore very surprised to observe cold responses in 728 of the 755 keratinocytes isolated from hibernating squirrels (4°C core body temperature). Upon warming, keratinocytes release ATP that subsequently activates nearby keratinocytes and/or sensory neurons via P2X receptor signaling (*Mandadi et al., 2009*; *Moehring et al., 2018*). If the same signaling is initiated by cooling, this high level of keratinocyte activity could lead to depolarization events in all sensory afferent terminals that express purinergic receptors, including those that co-express TRPM8. These depolarization events may lead to subthreshold membrane potential oscillations that encode temperature fluctuations over time. Alternatively, this cold-induced keratinocyte activity, and subsequent release of chemical mediators, could be a novel non-neuronal mechanism that allows

hibernating animals to detect changes in environmental temperature, a process that seems critical to the initiation and maintenance of hibernation.

In the ultimate translational test, cold responses were compared between keratinocytes isolated from glabrous rodent skin and hairy human skin. Over 67% of human keratinocytes responded to decreasing temperatures (*Figure 1B*);>50% of the keratinocytes from each of the five tissue donors were cold sensitive (*Figure 1—figure supplement 1A*). These results are similar to a previous report in which 60% of undifferentiated human keratinocytes responded to cold (*Tsutsumi et al., 2010*) (~24–12°C ramp). When compared to all of the rodent species, human keratinocyte cold responses were most similar to those observed in mouse. Human keratinocytes were more responsive to cold than rat keratinocytes, exhibiting larger cold-induced calcium transients that were initiated at higher temperatures. Similarly, human keratinocytes were less responsive to cold than squirrel keratinocytes; a higher percentage of squirrel keratinocytes responded to cold with larger calcium transients (*Figure 1*). Inter-sample variability in cold-response magnitude and temperature threshold were noted in the human samples, despite all keratinocytes being harvested from female breast skin (*Figure 1—figure supplement 1*). The average cold threshold in our human samples was ~19°C; in previous studies, cold thresholds were not reported, but peak cold responses were observed between 16–18°C (*Tsutsumi et al., 2010*). The rate of cooling, dictated by buffer flow rates and speed of heat exchange within the cooling device, or culturing conditions may explain these differences. Unfortunately, we do not understand the perceptual consequences of these species differences at this time since keratinocyte-specific manipulations are difficult to perform in any species besides mouse. However, the robust, conserved activity across all species tested suggests that mammalian keratinocytes play a larger functional role in thermosensation than simply buffering cold temperatures.

## Cold responses in mouse keratinocytes require release of intracellular calcium stores

We know very little about the molecular basis of keratinocyte cold transduction. To characterize the molecular cold transducers in mouse tissue, we performed calcium imaging on primary mouse keratinocytes (*Figure 2A*). Cold-induced calcium responses were first measured in keratinocytes incubated with 20 µM EGTA and 0 µM $Ca^{2+}$ to determine if extracellular calcium influx is required. Under these conditions, only 36% of keratinocytes responded to cold (*Figure 2B*); cells that did respond exhibited smaller calcium transients (25% decrease from vehicle; *Figure 2C*) and responded at lower temperatures (0.6°C decrease from vehicle; *Figure 2D*) than vehicle-exposed cells. Removal of extracellular sodium (equimolar NMDG substitution) had no additional effect on keratinocyte cold responses, therefore arguing against cold-sensitive sodium channels (e.g. epithelial sodium channel (ENaC) [*Askwith et al., 2001*]) being the primary cold detectors. Keratinocyte cold responses were only completely eliminated when the extracellular buffer included 20 µM EGTA and 0 µM $Ca^{2+}$ and intracellular calcium stores were depleted with 1 µM thapsigargin immediately prior to the cold ramp (*Figure 2B*).

Based on these experiments, there are two scenarios to explain how keratinocytes detect cold: 1) the primary cold sensor is expressed in the endoplasmic reticulum and upon buffer cooling, cytosolic calcium is increased via activation of calcium release-activated channels (CRAC) like the STIM/Orai complex or plasma-membrane expressed voltage-gated channels or 2) there are one or more cold sensors expressed in the endoplasmic reticulum and additional cold sensors expressed in the plasma membrane. To address the former, keratinocytes were incubated with 5J4, a CRAC channel blocker. While not a member of the TRP superfamily, the STIM/Orai CRAC channels are required for normal warming and 'heat-off' responses in human and mouse keratinocytes (*Liu et al., 2019*). This 'heat-off' response (i.e. decrease in temperature following removal of a heat stimulus) could be classified as cooling, and while not explicitly related to discrete 'cold' temperatures, may suggest a role for STIM/Orai signaling in keratinocyte cold responses. However, incubation with 5J4 did not change the percentage of keratinocytes responding to cold (*Figure 2B*) or the magnitude of the peak cold response (*Figure 2C*). Extracellular application of 5J4 did lower the average cold threshold by 0.4°C, but the biological significance of this decreases is unclear (*Figure 2D*). Bidaux et al. identified a novel epidermal isoform of the cold and menthol-sensitive (*McKemy et al., 2002*; *Peier et al., 2002a*) channel TRPM8 (eTRPM8) in the endoplasmic reticulum of human keratinocytes (*Bidaux et al., 2015*). Based on expression patterns, this channel could be the intracellular sensor that is mediating cold responses in calcium-free buffer. However unlike *Bidaux et al., 2015*, we did

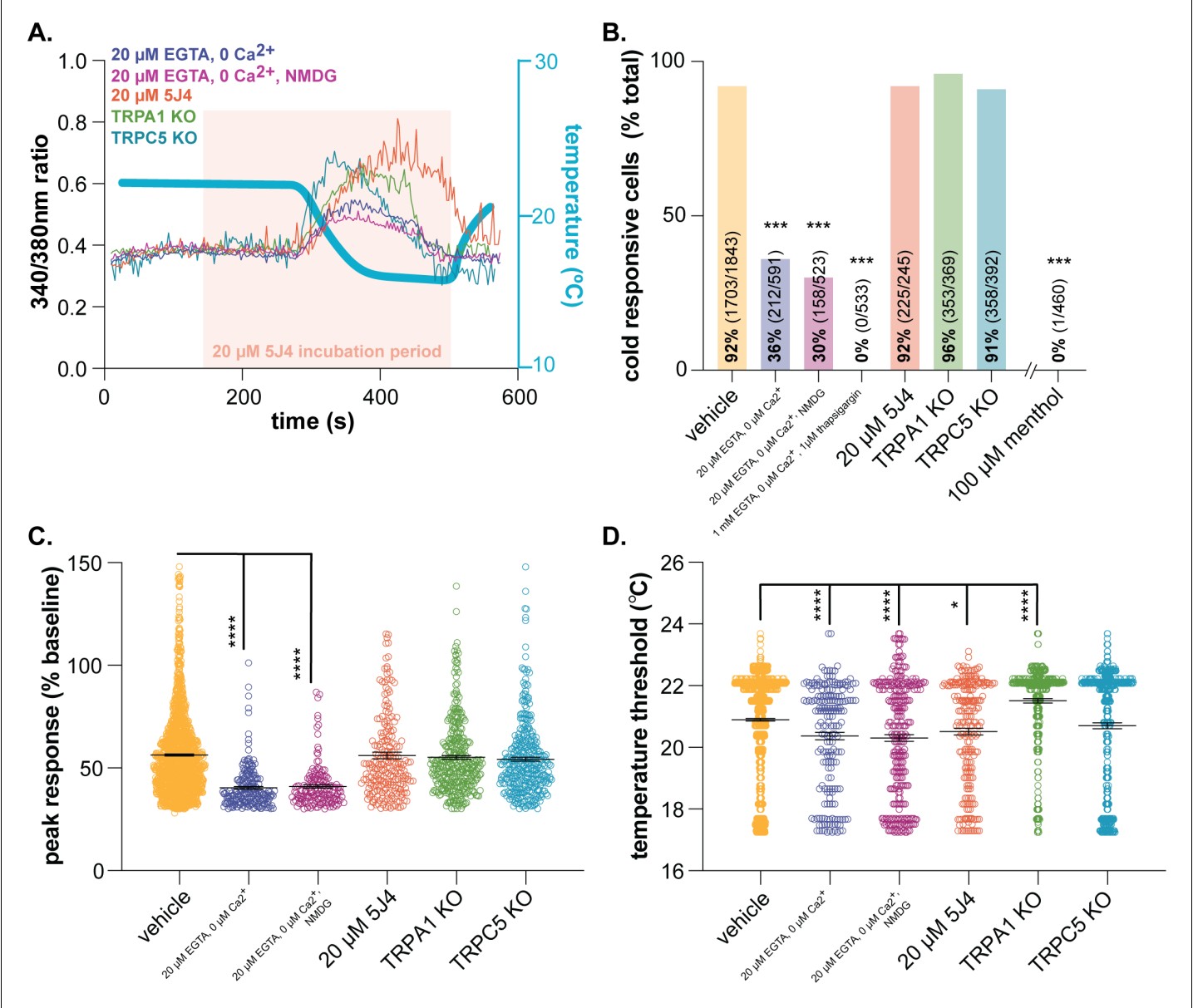

**Figure 2.** Characterizing murine cold sensor(s) in keratinocytes. (**A**).) In order to characterize the proteins involved in cold transduction, cold-induced calcium transients were measured in keratinocytes exposed to extracellular buffer containing different ionic concentrations or pharmacological agents, or in keratinocytes from transgenic mice. (**B**).) Extracellular calcium chelation (EGTA, 0 µM $Ca^{2+}$) decreases the proportion of keratinocytes that respond to cold; substituting NMDG for extracellular sodium did not further decrease the percentage of cold-responsive cells. Cold responses were only abolished when endoplasmic reticulum $Ca^{2+}$ stores were depleted and unable to be refilled with extracellular sources (EGTA, 0 µM $Ca^{2+}$, thapsigargin). CRAC channel inhibition (5J4) did not decrease the percentage of cold-responsive cells, and similar proportions of keratinocytes from C57BL/6, global TRPA1, and global TRPC5 knockout mice responded to decreasing buffer temperature (Chi square $p < 0.0001$; Fisher's Exact tests ***$p < 0.001$ vs. vehicle; $n \geq 4$). (**C**).) Peak calcium responses to cold were lower in the absence of extracellular calcium; responses were unaltered in other conditions (1-way ANOVA $p < 0.0001$; Bonferroni's multiple comparisons: ****$p < 0.0001$ vs. vehicle, #### $p < 0.0001$ vs. EGTA, 0 µM $Ca^{2+}$, &&&& $p < 0.0001$ vs. EGTA, 0 µM $Ca^{2+}$, NMDG). (**D**).) Altering extracellular buffer contents increased or decreased the temperature at which keratinocytes responded to cold by $\leq 0.6°C$. Mean temperature thresholds (°C) for vehicle: 20.9, EGTA, 0 µM $Ca^{2+}$,±NMDG: 20.3, 20 µM 5J4: 20.5, TRPA1 KO: 21.5, TRPC5 KO: 20.7 (Kruskal-Wallis test $p < 0.0001$; Dunn's multiple comparisons: *$p < 0.05$, ****$p < 0.0001$ vs. vehicle).

not observe menthol-induced calcium transients in mouse keratinocytes (*Figure 2B*). Given that Biduax et al. performed their experiments in conditions that reduced capacitive calcium flux (i.e. reduced CRAC channel activity), we expected to observe even larger menthol responses in our experiments, but only 1 of the 460 mouse keratinocytes tested responded to menthol; Biduax et al.

did not report the percentage of menthol-sensitive keratinocytes in their studies. Regardless of these discrepancies in menthol-sensitivity, there is essentially no published evidence to support that eTRPM8 is the primary keratinocyte cold sensor. The original eTRPM8 publication did not determine whether keratinocytes from TRPM8 knockout mice exhibited regular responses to buffer cooling, or if a mutation in this channel altered cold-induced activity in human keratinocytes (*Bidaux et al., 2015*). Additionally, heterologous cells expressing eTRPM8 did not exhibit whole-cell currents during extracellular menthol application or temperature decreases (*Bidaux et al., 2015*). Based on these data and our current experiments, we do not believe that eTRPM8 is the primary intracellular cold sensor in mouse keratinocytes.

To characterize potential plasma membrane cold sensors, keratinocytes were isolated from global TRPA1 and TRPC5 knockout mice. TRPA1 is a member of the TRP ion channel superfamily involved in detection of noxious cold (<17°C) (*Story et al., 2003*). In human keratinocyte studies, extracellular application of the TRPA1 antagonist HC-030031 partially decreases cold-induced calcium transients (*Tsutsumi et al., 2011*). In mouse, the percentage of cold-sensitive cells and peak cold responses in TRPA1 knockout keratinocytes were similar to those observed in wildtype keratinocytes (*Figure 2B and C*). On average, TRPA1 knockout keratinocytes exhibited cold-induced calcium transients at warmer temperatures than wildtype keratinocytes (21.5°C vs. 20.9°C; *Figure 2D*). This lack of TRPA1 involvement was not surprising as we and others have not been able to verify TRPA1 expression or function in mouse epidermal cells (*Bouvier et al., 2018*; *Liu et al., 2013*; *Zappia et al., 2016*). Additionally, global TRPA1 knockout mice (*Bautista et al., 2006*) and rats (*Reese et al., 2020*) exhibit normal cold-induced reflexive behaviors, further suggesting that this channel is not a conserved noxious cold transducer in mammalian skin cells. TRPC5 is another cold-sensitive TRP channel family member (*Zimmermann et al., 2011*) expressed in keratinocytes (*Beck et al., 2006*) that could be a plasma membrane-expressed cold sensor. TRPC5 knockout keratinocyte cold responses were identical to those observed in wildtype tissue (*Figure 2*). This result may be related to the temperatures used in this experiment; when expressed in heterologous cells, TRPC5 currents peak around 25°C (*Zimmermann et al., 2011*), the starting temperature of our experiments. Alternatively, TRPC5 activity may not be required for normal cold sensation; global TRPC5 knockout mice exhibit normal cold behavioral responses (*Zimmermann et al., 2011*). Identification and validation of *novel* cold-sensing proteins is a time-intensive process that is beyond the scope of this manuscript. Based on our characterization studies however, there is at least one intracellular cold-sensitive protein that is required for keratinocyte cold activity. Clues from similar characterization studies and comparative genomics could be used to identify putative cold sensors that are differentially expressed in keratinocytes from the species tested above.

## Optogenetic inhibition of keratinocytes decreases behavioral cold sensitivity

After successfully demonstrating that mammalian keratinocytes are cold-responsive, we used the same global inhibitory optogenetic approach from our previous study (*Moehring et al., 2018*) to determine whether keratinocytes are required for normal cold sensation. We generated transgenic mice expressing Archaerhodopsin, a light-sensitive proton pump, in keratin14-positive cells (keratinocytes and Merkel cells; Krt14:Arch Cre+) and Cre- littermate controls (Krt14:Arch Cre-). To determine if optogenetic inhibition of keratinocytes affects in vivo cold sensitivity, the cold plantar assay (*Brenner et al., 2012*) was performed in Krt14:Arch Cre+ and Krt14:Arch Cre- mice. The cold plantar assay is our preferred measure of cold sensitivity since behavioral responses are highly reproducible and easy to quantify. The nature of the dry ice stimulus elicits anticipatory cold-related behaviors rather than pain-related behaviors elicited by discrete noxious cold temperatures. On average, animals respond when the paw temperature has dropped by ~2°C, a change that is well within the innocuous cool range (*Brenner et al., 2012*). Krt14:Arch Cre+ mice exhibited longer withdrawal latencies to the dry ice stimulus when 590 nm light was applied to the stimulated hindpaw (*Figure 3A*). Based on these data, keratinocyte activity contributes to normal cold sensation and behavioral responses in vivo.

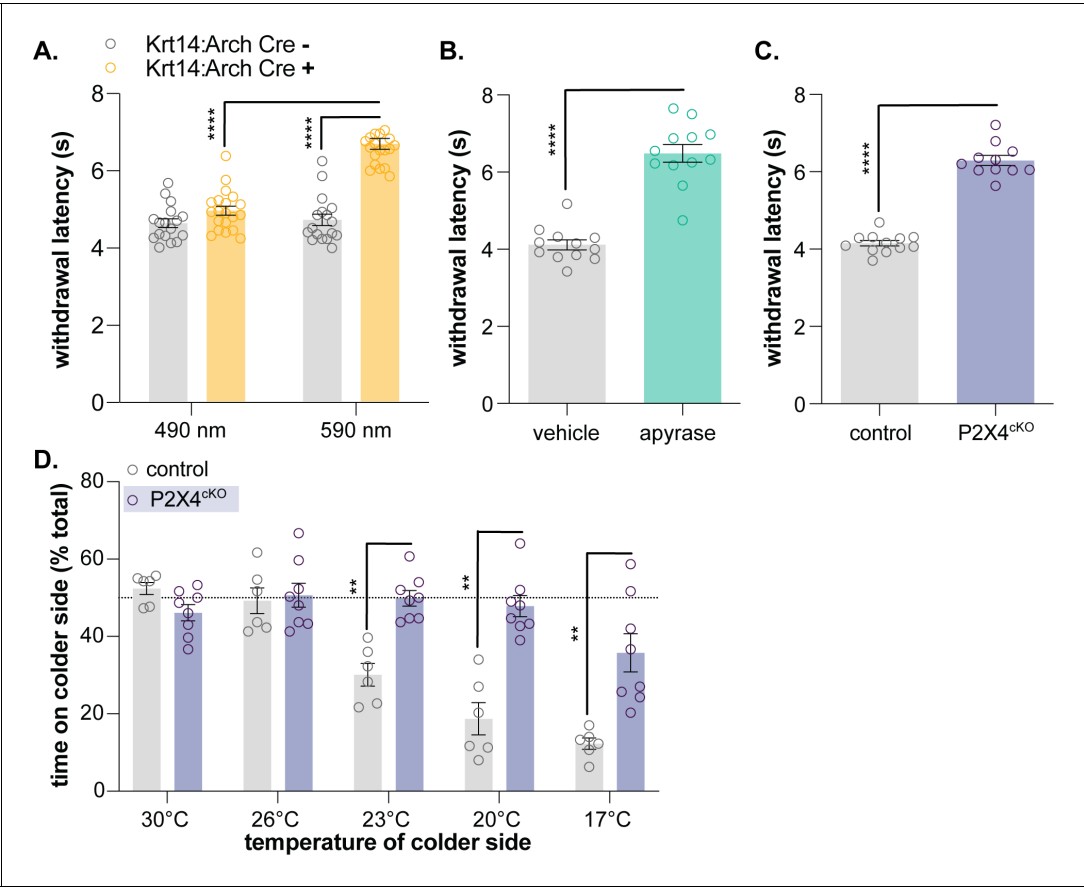

**Figure 3.** Keratinocyte-to-sensory neuron signaling is required for normal cold sensation. (**A**) 590 nm light exposure increases the withdrawal latency of Krt14:Arch Cre+ mice during plantar dry ice stimulation (2-way ANOVA main effects of light, genotype, and light x genotype interaction p<0.0001; Bonferroni's multiple comparisons: ****p<0.0001 Krt14:Arch Cre+ 490 vs. 590 nm, 590 nm Krt14:Arch Cre+ vs. Cre-; n = 17–20) (**B**) Intraplantar administration of apyrase (0.4 units; catalyzes ATP hydrolysis) increases the withdrawal latency of C57BL/6 mice during plantar dry ice stimulation (unpaired t-test ****p<0.0001; n = 12). (**C**) Animals lacking sensory neuron P2X4 receptors (P2X4cKO) exhibit increased withdrawal latencies to plantar dry ice stimulation (unpaired t-test ****p<0.0001; n = 11–12). (**D**) In a two-temperature preference test, P2X4cKO mice spend more time on innocuous cold surfaces than their P2X4 expressing littermates (2-way ANOVA main effects of temperature, genotype, and temperature x genotype interaction p<0.0001; Bonferroni's multiple comparisons: **p<0.01 P2X4cKO vs. control; n = 6–8).

## Purinergic signaling is required for normal cold sensation

We previously demonstrated that epidermal ATP signaling is required for normal mechanosensation (*Moehring et al., 2018*). We and others have shown that keratinocytes release ATP when mechanically stimulated (*Klusch et al., 2013*; *Koizumi et al., 2004*; *Moehring et al., 2018*). This ATP can subsequently bind to members of the P2Y receptor family or, as demonstrated in our previous publication, P2X4 receptors expressed in Aβ, Aδ, and C fiber afferents, effectively depolarizing neuronal membranes and increasing the likelihood of fiber firing (*Koizumi et al., 2004*; *Moehring et al., 2018*). To determine if epidermal purinergic signaling is similarly required for cold sensation, we performed the cold plantar assay on wildtype mice following intraplantar injection of apyrase, an enzyme that catalyzes ATP hydrolysis. Apyrase-injected animals exhibited longer withdrawal latencies to the dry ice stimulus than vehicle-injected controls (*Figure 3B*). We also tested the cold sensitivity of sensory neuron-specific P2X4 knockout mice (P2X4cKO; AdvilCre::P2rX4 fl/fl). P2X4cKO mice exhibited longer withdrawal latencies to dry ice stimulation than control littermates (*Figure 3C*). To complement this reflexive cold behavioral measure, thermotactic behaviors were assessed in P2X4cKO mice using a two-temperature preference test. In this assay, animals choose to spend time on a floor held at 30°C or a floor that is held at a colder temperature. Control mice displayed a place preference for the 30°C floor when it was paired with floors held at 23, 20, and 17°C; P2X4cKO mice spent more time on these colder floors than controls (*Figure 3D*). Collectively, these findings are

similar to those observed in mechanosensory tests and suggest that ATP-P2X4 signaling also mediates normal cold sensation in vivo.

## Purinergic keratinocyte signaling contributes to normal heat sensation

Since the noxious heat receptor TRPV1 was visualized in human keratinocytes (*Denda et al., 2001*), many have hypothesized that these non-neuronal cells are involved in thermosensation (*Chung et al., 2004*; *Dhaka et al., 2006*; *Lee and Caterina, 2005*; *Moqrich, 2005*; *Peier et al., 2002b*). Primary human and mouse keratinocytes exhibit heat-evoked currents (*Chung et al., 2004*; *Liu et al., 2019*; *Tsutsumi et al., 2011*) and release ATP upon warming (*Mandadi et al., 2009*). These heat responses are reportedly mediated by TRPV3 (*Moqrich, 2005*), TRPV4 (*Chung et al., 2004*), and STIM1/Orai channels (*Liu et al., 2019*), but global TPRV3/TRPV4 double knockout mice exhibit normal behavioral responses to innocuous and noxious heat stimulation (*Huang et al., 2011*), and keratinocytes isolated from these animals exhibit heat responses identical to those observed in wildtype tissues (*Liu et al., 2019*). Furthermore, we and others have not observed capsaicin responses in primary mouse keratinocytes (*Chung et al., 2004*), suggesting that there may be species-specific differences in TRPV1 keratinocyte expression, similar to TRPA1. Regardless, it is highly unlikely that knockdown of one or more temperature-sensitive channels will reveal the full extent to which keratinocytes are required for heat sensation. Similar to sensory neurons (*Vandewauw et al., 2018*), keratinocytes may express a complement of functionally redundant proteins that compensate for reduced activity of related temperature-sensitive channels. Therefore, in order to determine how this heterogeneous cell population contributes to naïve heat sensation, global inhibitory approaches are required. Baumbauer, DeBerry, and Adelman et al. previously demonstrated that keratinocyte activity contributes to ex vivo peripheral thermal transduction processes; optogenetic inhibition of keratin14-expressing cells decreased activity in heat-sensitive polymodal C and Aδ nociceptors (*Baumbauer et al., 2015*). To determine if keratinocyte inhibition affects in vivo heat sensitivity, we performed the Hargreaves radiant heat assay (*Hargreaves et al., 1988*) on Krt14:Arch Cre+ and Krt14:Arch Cre- mice. Krt14:Arch Cre+ mice exhibited longer withdrawal latencies to the heat stimulus when 590 nm light was applied to the stimulated hindpaw (*Figure 4A*). Wildtype animals injected with apyrase and P2X4cKO mice also exhibited longer withdrawal latencies to the heat stimulus when

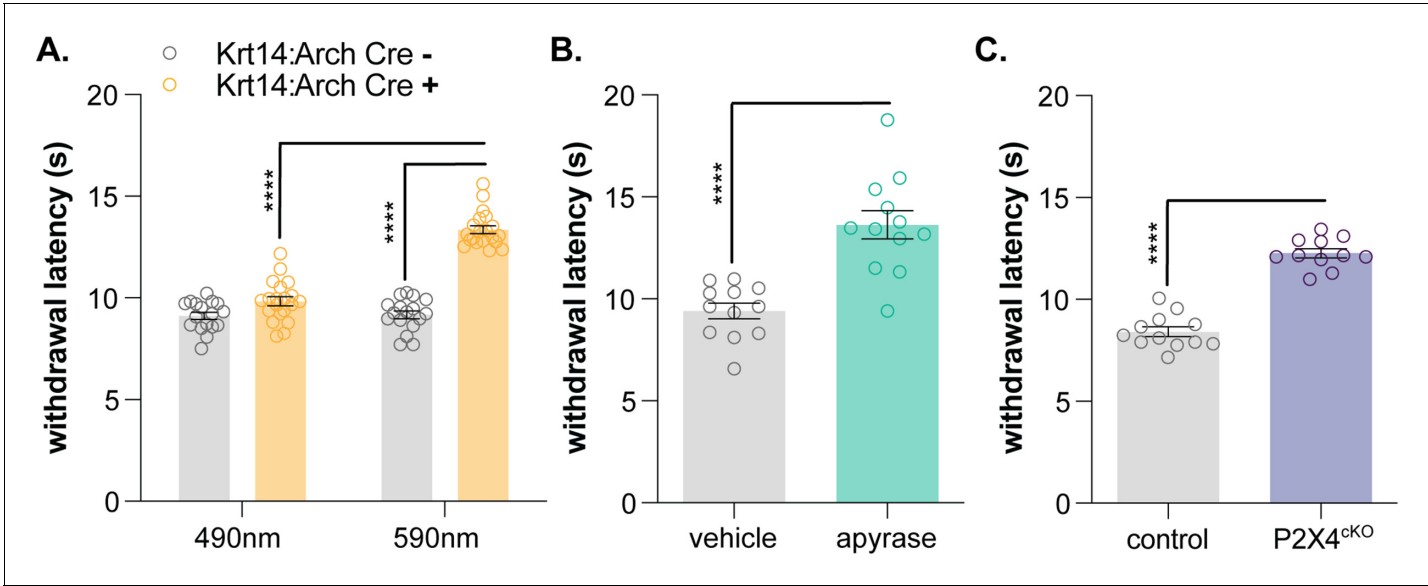

**Figure 4.** Keratinocyte-to-sensory neuron signaling is required for normal heat sensation. (**A**) 590 nm light exposure increases the withdrawal latency of Krt14:Arch Cre+ mice during plantar radiant heat stimulation (2-way ANOVA main effects of light, genotype, and light x genotype interaction p<0.0001; Bonferroni's multiple comparisons: ****p<0.0001 Krt14:Arch Cre+ 490 vs. 590 nm, 590 nm Krt14:Arch Cre+ vs. Cre-; n = 17–20). (**B**) Intraplantar administration of apyrase (0.4 units; catalyzes ATP hydrolysis) increases the withdrawal latency of C57BL/6 mice during plantar radiant stimulation (unpaired t-test ****p<0.0001; n = 10). (**C**) Animals lacking sensory neuron P2X4 receptors (P2X4cKO) exhibit increased withdrawal latencies to plantar radiant heat stimulation (unpaired t-test ****p<0.0001; n = 11–12).

compared to the appropriate controls (*Figure 4B and C*). Collectively, these and our previous data (*Moehring et al., 2018*) are the first to demonstrate the conserved requirement of keratinocyte activity for normal mechanical, cold, and heat sensation in vivo. While it is true that primary sensory neurons are capable of detecting and encoding environmental stimuli without accessory cells, our work suggests that proper somatosensation also requires critical stimulus amplification at the level of the keratinocyte. Keratinocyte population activity (*Koizumi et al., 2004*) could also explain the extreme variability observed in DRG cold responses in vivo (*Luiz et al., 2019*; individual sensory neuron activity could be directly influenced by the non-neuronal cells in the local peripheral terminal environment. Before our work, global chemo- (*Pang et al., 2015*) and optogenetic (*Baumbauer et al., 2015*; *Moehring et al., 2018*) approaches had been used to illustrate the importance of keratinocyte-to-sensory neuron signaling in somatosensation. However, these non-specific activation methods did not allow for the interrogation of specific sensory modalities as generic keratinocyte activation quickly induced nocifensive and paresthesia-like behaviors in mice (*Baumbauer et al., 2015*; *Moehring et al., 2018*; *Pang et al., 2015*). These and our former studies (*Moehring et al., 2018*) also position epidermal purinergic signaling as a ubiquitous but critical mechanism that amplifies many modalities of somatosensory stimuli. Additional studies are required to identify modality-specific signaling pathways and keratinocyte-secreted factors that can be exploited for topical therapeutic development for pain and itch conditions.

## Materials and methods

### Key resources table

| Reagent type (species) or resource | Designation | Source or reference | Identifiers | Additional information |
|---|---|---|---|---|
| Strain, strain background *Rattus norvegicus* | Sprague Dawley rat | Taconic Biosciences | NTac:SD; SD-F, SD-M | |
| Strain, strain background *Ictidomys tridecemlineatus* | 13-lined ground squirrel | University of Wisconsin Oshkosh Squirrel Colony | | |
| Strain, strain background *Mus musculus* | Krt14:Arch Cre+ Krt14:Arch Cre- | *Moehring et al., 2018* | K14:Arch Cre+ K14:Arch Cre- | |
| Strain, strain background *Mus musculus* | P2X4$^{cKO}$ | *Moehring et al., 2018* | *Advil*$^{Cre+}$::*P2rX4*$^{fl/f}$ *Advil*$^{Cre-}$::*P2rX4*$^{fl/f}$ | |
| Strain, strain background *Mus musculus* | TRPA1 KO | *Kwan et al., 2006* | | |
| Strain, strain background *Mus musculus* | TRPC5 KO | The Jackson Laboratory | Trpc5$^{tm1.1Lbi}$/Mmjax Jackson Stock: 37349-JAX | |
| Other | In-line heater/cooler | Warner Instruments | https://www.warneronline.com/in-line-heater-cooler-sc-20 | |
| Other | Refrigerated circulator | Julabo | https://www.julabo.com/en-us/products/refrigerated-circulators/refrigerated-heating-circulators/f25-he | |
| Other | USB thermo couple probe | | https://www.omega.com/en-us/sensors-and-sensing-equipment/temperature/sensors/thermocouple-probes/p/TJ-USB | |

Most of the methods used in this manuscript are identical to those used in our previous publication (*Moehring et al., 2018*; methods unique to this manuscript are described below.

## Animals

All protocols were in accordance with National Institutes of Health guidelines and were approved by the Institutional Animal Care and Use Committee at the Medical College of Wisconsin (Milwaukee, WI; protocol #383). Keratinocytes were isolated from bilateral glabrous hindpaw skin of mixed strains of mice, (mixed males and females, aged 8–16 weeks, bred in house), Sprague Dawley rats (2 males and two females, aged 10 weeks from Taconic Biosciences) and 13-lined ground squirrels (four females, aged 33 weeks from University of Wisconsin Oshkosh Squirrel Colony; hibernating at 4°C when sacrificed). Glabrous hindpaw skin was excised and immediately incubated in room temperature Hanks Balanced Salt Solution (without $CaCl_2$, $MgCl_2$, or $MgSO_4$), supplemented with 10 mg/mL dispase. Note, hibernating ground squirrel skin was dissected and processed at room temperature in an identical manner to mouse and rat skin. After 45 min, the epidermis was peeled from the dermis and keratinocyte culture was performed as described below. Data from individual sexes were analyzed separately then combined since no differences were recorded.

## Human tissue

Keratinocytes were obtained from breast skin tissue collected by the Medical College of Wisconsin Tissue Bank (five female donors aged 41, 50, 54, 52, and 49). After collection, skin samples were stored in 4°C Hanks Balanced Salt Solution (without $CaCl_2$, $MgCl_2$, or $MgSO_4$), supplemented with 10 mg/mL dispase. All samples were collected by members of the Stucky Lab within 12 hr of dissection and stored at 4°C overnight. The following day, the epidermis was peeled from the dermis and keratinocyte culturing was performed as described below.

## Primary keratinocyte isolation and culture

Human and rodent epidermal sheets were incubated in plastic Petri dishes containing 50% EDTA and 0.05% trypsin in Hanks Balanced Salt Solution (without $CaCl_2$, $MgCl_2$, or $MgSO_4$) for 27 min at room temperature. After addition of heat-inactivated FBS (15%), epidermal sheets were rubbed vigorously (>1 min) against the bottom of the Petri dish using a disposable pipette tip to dissociate the tissue. Following this step, the buffer was always quite cloudy and contained 'sticky' strings of tissue (i.e. dissociated keratinocytes); cloudiness depended on starting material volume and dissociation times. The buffer was transferred to a clean tube and centrifuged, after which the supernatant was removed. The keratinocyte-containing pellet was re-suspended and grown in EpiLife medium supplemented with 1% human keratinocyte growth supplement, 0.2% Gibco Amphotericin B, and 0.25% penicillin-streptomycin. Cells were grown on laminin-coated coverslips at 37°C and 5% $CO_2$. Medium was exchanged 24 hr after plating and, if required, every 48 hr thereafter. Mouse, rat, and squirrel keratinocytes were imaged 3 days after plating. At this timepoint, all rodent cultures were essentially a confluent monolayer of proliferative, undifferentiated cells. Human keratinocytes were imaged when the same culture morphology was observed (typically 4–5 days following plating).

## Calcium imaging

Calcium imaging was performed on keratinocytes using the dual-wavelength ratiometric calcium indicator dye Fura-2-AM. Keratinocytes were loaded with 2.5 µg/mL Fura-2 in 2% BSA for 45 min at room temperature then washed with extracellular buffer for 30 min. Extracellular buffer (pH 7.4, 320 Osm) contained (in mM): 150 NaCl, 10 HEPES, 8 glucose, 5.6 KCl, 2 $CaCl_2$, and 1 $MgCl_2$. Coverslips were mounted onto an inverted fluorescence microscope and perfused with room temperature (~24°C) extracellular buffer at a rate of 6 mL/min. Buffer temperature was detected by a thermocouple temperature probe and recorded in LabChart software throughout the experiment. To generate a cold ramp, extracellular buffer was cooled in real time through an in-line cooler (water jacket perfused with antifreeze chilled via refrigerated circulator); temperatures dropped to ~12°C over 2 min. Fluorescence images were captured at 340 and 380 nm. Responsive cells were those that exhibited >30% increase in 340/380 nm ratio from baseline during buffer cooling; cold thresholds were defined as the bath temperature at which the 340/380 nm ratio was first >30% baseline. 5J4 (100 µM), menthol (100 µM), and thapsigargin (1 µM) were dissolved in 1% DMSO. In EGTA (20 µM) and NMDG (150 mM) experiments, mannitol was added to keep solution osmolarity identical to that of normal extracellular buffer (320 Osm). Keratinocytes were randomized to drug treatment groups.

The experimenter was blinded to species, genotype, and/or treatment until data analysis was complete.

## Behavioral testing

The cold plantar assay (*Brenner et al., 2012*) was used as a reflexive measure of cold sensitivity. Animals were placed into behavior chambers on 2.5 mm thick glass surface for >1 hr of habituation. Powdered dry ice was packed into a 10 mL syringe, then applied to the glass beneath the plantar surface of the hindpaw. Withdrawal latencies were measured three times for each paw and then averaged. For optogenetic manipulation, a 590 nm LED or 490 nm control LED was placed ~3 cm below the glass surface for 1 min prior to and during stimulus application. At this distance, the power of the 590 nm LED was 17.5 mW and the power of the 490 nm LED was 21.6 mW. The two-temperature preference test was used to measure thermotactic behaviors (*Moqrich, 2005*). Each side of the testing apparatus contains a thermally regulated metal plate that was set to a specific temperature. Animals were habituated to the apparatus (both floors set to 30℃) for 15 min prior to the testing day. On testing day, both floors were set to 30℃ for 5 min baseline test. After baseline testing, the temperature of one floor was dropped to 26℃, 23℃, 20℃, or 17℃ and animal place preferences were recorded for 5 min. No more than three temperature tests were performed per animal per day. To control for inherent preference of the sides, we randomly altered both the plate where the animals were introduced and the temperatures on each plate. The percent time spent on the colder plate was calculated. The Hargreaves assay (*Hargreaves et al., 1988*) was used as a reflexive measure of heat sensitivity. Animals were placed into behavioral chambers on glass plate for >1 hr of habituation. A focal radiant heat source was applied to the glass beneath the plantar surface of the hindpaw and withdrawal latencies were recorded. Withdrawal latencies were measured three times for each paw then averaged. For optogenetic manipulation, a 590 nm LED or 490 nm control LED was placed ~3 cm below the glass surface for 1 min prior to and during stimulus application. Apyrase treatment (described in *Moehring et al., 2018*) occurred 45 min prior to cold or heat testing. The experimenter was blinded to genotype and/or treatment until data analysis was complete.

## Acknowledgements

We would like to thank Dr. Ben Sajdak, Alex Salmon, Hannah Follett, and Dr. Joe Carroll for providing us with 13-lined ground squirrel tissue and Alex Mikesell, Dr. Jonathan Marchant, Dr. Quinn Hogan, Dr. Wai-Meng Kwok, and Dr. Jon Sack for helpful discussions about data interpretations and reagent access.

## Additional information

### Funding

| Funder | Grant reference number | Author |
|---|---|---|
| National Institutes of Health | NS040538 | Cheryl L Stucky |
| National Institutes of Health | NS070711 | Cheryl L Stucky |
| National Institutes of Health | NS108278 | Cheryl L Stucky |
| National Institutes of Health | NS106789 | Katelyn E Sadler |
| Advancing a Healthier Wisconsin Endowment, Medical College of Wisconsin | | Cheryl L Stucky |

The funders had no role in study design, data collection and interpretation, or the decision to submit the work for publication.

### Author contributions

Katelyn E Sadler, Conceptualization, Data curation, Formal analysis, Funding acquisition, Investigation, Visualization, Methodology, Writing - original draft, Writing - review and editing; Francie

Moehring, Conceptualization, Data curation, Formal analysis, Investigation, Visualization, Methodology, Writing - review and editing; Cheryl L Stucky, Conceptualization, Supervision, Funding acquisition, Writing - review and editing

### Author ORCIDs
Katelyn E Sadler https://orcid.org/0000-0003-2078-3527
Francie Moehring https://orcid.org/0000-0002-0071-5685
Cheryl L Stucky https://orcid.org/0000-0003-4966-6594

### Ethics
Animal experimentation: All protocols were in accordance with National Institutes of Health guidelines and were approved by the Institutional Animal Care and Use Committee at the Medical College of Wisconsin (Milwaukee, WI; protocol #383).

### Decision letter and Author response
Decision letter https://doi.org/10.7554/eLife.58625.sa1
Author response https://doi.org/10.7554/eLife.58625.sa2

## Additional files

### Supplementary files
- Source data 1. Figure individual data points.
- Transparent reporting form

### Data availability
All data generated or analyzed during this study are included in the manuscript and supporting files.

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
