## [Decision Letter]

**Acceptance summary:**

This study make extensive use of pharmacological, knockout and optogenetic approaches to demonstrate keratinocyte responsiveness to cold, and contributions of keratinocytes to behavioral responses to heat and cold stimuli. Well-designed experiments include an interesting comparative analysis of cold responses across species. The manuscript extends our understanding of keratinocyte contributions to somatosensory transduction and is appropriate for publication in eLife.

**Decision letter after peer review:**

Thank you for submitting your article "Keratinocytes are required for normal cold and heat sensation" for consideration by *eLife*. Your article has been reviewed by three peer reviewers, one of whom is a member of our Board of Reviewing Editors, and the evaluation has been overseen by Richard Aldrich as the Senior Editor. The following individuals involved in review of your submission have agreed to reveal their identity: Kathryn M Albers (Reviewer #2); Derek Moliver (Reviewer #3).

The reviewers have discussed the reviews with one another and the Reviewing Editor has drafted this decision to help you prepare a revised submission.

Summary:

This manuscript extends a previous paper by the authors demonstrating that keratinocytes are mechanically sensitive and contribute to somatosensory mechanical transduction. Here, the authors make extensive use of pharmacological, knockout and optogenetic approaches to demonstrate keratinocyte responsiveness to cold, and contributions of keratinocytes to behavioral responses to heat and cold stimuli. Well-designed experiments include an interesting comparative analysis of cold responses across species, although the physiological significance of species differences remains to be resolved. Sex differences were evaluated and not observed. Several candidate transducers from the literature were tested for contributions to the keratinocyte cold responses and excluded (eTRPM8, TRPA1, TRPC5). Generation of cold-induced calcium transients required release of calcium from intracellular stores, leading the authors to conclude that there may be an intracellular cold transducer. Purinergic communication from keratinocytes to sensory axons was proposed based on attenuated behavioral responses with intradermal ATP depletion and sensory neuron-specific knockout of P2X4, following up on results from the previous study that identified a role for P2X4-mediated purinergic transmission between keratinocytes and sensory neurons in mechanical transduction. The manuscript extends our understanding of keratinocyte contributions to somatosensory transduction and is appropriate for publication in *eLife*.

The following items should be addressed.

Essential revisions:

1) In the voltage clamp experiments shown in Figure 3, keratinocytes showed an outward current following cold stimulation. This is counterintuitive, since an outward current may not "activate" the cell, which conflicts with other findings in the study. This issue should be addressed (and the composition of internal solution used for patch clamp recordings should be described).

Related to this, the basis of optogenetic "inhibition" comes from the finding that expressing archaerhodopsin in keratinocytes reduces the outward current density, so there would be less hyperpolarization upon illumination with 590 nm light. This is also counterintuitive and needs further clarification.

Although the findings argue that intracellular calcium stores contribute to the calcium signal upon cold stimulation, the functional experiments rely on manipulating membrane currents for optogenetic inhibition, which as mentioned wasn't well resolved by the voltage-clamp experiments. Also, as far as I know, a rise in intracellular calcium doesn't usually occur with plasma membrane hyperpolarization, as indicated by the outward currents. This seems difficult to explain unless there are calcium-activated potassium channels or chloride channels at play here.

Therefore, with regards to the patch clamp experiments and membrane current measurements, there is some confusion. In order to clarify, we suggest the following experiments:

a) For voltage clamp measurements, hold cells at the resting potential during cold stimulation and assess the response.

b) Current-clamp measurements would help determine how cold temperatures and optogenetic inhibition during cold stimulation lead to a change in membrane voltage.

2) The title of this submission, 'Keratinocytes are required for normal cold and heat sensation' is overstating the findings. A possible revision is 'Keratinocytes contribute to normal cold and heat sensation'. The only evidence that supports a requirement for keratinocytes is from the behavioral measures of cold responses in the presence of Arch activation where a slight increase in latency to cold is measured (Figure 3). Although supportive, until it is shown that alteration (inhibition) of keratinocytes elicits a change in response properties of cutaneous (cold sensing) neurons, the actual impact of keratinocytes on neural transmission of cold stimuli can be debated.

---

## [Author Response]

Essential revisions:1) In the voltage clamp experiments shown in Figure 3, keratinocytes showed an outward current following cold stimulation. This is counterintuitive, since an outward current may not "activate" the cell, which conflicts with other findings in the study. This issue should be addressed (and the composition of internal solution used for patch clamp recordings should be described).Related to this, the basis of optogenetic "inhibition" comes from the finding that expressing archaerhodopsin in keratinocytes reduces the outward current density, so there would be less hyperpolarization upon illumination with 590 nm light. This is also counterintuitive and needs further clarification.Although the findings argue that intracellular calcium stores contribute to the calcium signal upon cold stimulation, the functional experiments rely on manipulating membrane currents for optogenetic inhibition, which as mentioned wasn't well resolved by the voltage-clamp experiments. Also, as far as I know, a rise in intracellular calcium doesn't usually occur with plasma membrane hyperpolarization, as indicated by the outward currents. This seems difficult to explain unless there are calcium-activated potassium channels or chloride channels at play here.Therefore, with regards to the patch clamp experiments and membrane current measurements, there is some confusion. In order to clarify, we suggest the following experiments:a) For voltage clamp measurements, hold cells at the resting potential during cold stimulation and assess the response.b) Current-clamp measurements would help determine how cold temperatures and optogenetic inhibition during cold stimulation lead to a change in membrane voltage.

We completely agree with the reviewers; these data are counterintuitive and confusing. To better understand what could be happening in these recordings, we consulted with Dr. Jon Sack, a biophysicist with over 20 years of experience in patch clamp electrophysiology. We provided Dr. Sack with our original patch clamp files and he found several issues. Similar to the reviewers, Dr. Sack noted that our holding membrane potential was incorrect; many cells were being held at a value that was too negative. Culture variability or repeated buffer cooling (i.e. cells on a single cover slip were exposed to multiple cold ramps) may have altered these parameters from our previous study. Dr. Sack also observed high variability in our leak currents, and mismatching between series and pipette resistance values. Leak current variability may have resulted from multiple cooling events; repeated direct exposure to 12°C buffer may have destabilized cell membranes to the point of cells taking on an “injured” phenotype. Based on these observations, we no longer feel comfortable including these experiments in our manuscript. Therefore, the updated version of the manuscript does not include any patch clamping data. We would like to repeat these experiments and add in those suggested by the reviewers, but, in complete transparency, are unable to do so at this time since none of our current trainees can proficiently patch clamp keratinocytes. The COVID-19 pandemic and continued institutional access restrictions make the timeline for training and successful experimental completion even more uncertain. We hope that the reviewers understand this predicament and still deem the calcium imaging and in vivo experiments worthwhile to publish as a Research Advance.

2) The title of this submission, 'Keratinocytes are required for normal cold and heat sensation' is overstating the findings. A possible revision is 'Keratinocytes contribute to normal cold and heat sensation'. The only evidence that supports a requirement for keratinocytes is from the behavioral measures of cold responses in the presence of Arch activation where a slight increase in latency to cold is measured (Figure 3). Although supportive, until it is shown that alteration (inhibition) of keratinocytes elicits a change in response properties of cutaneous (cold sensing) neurons, the actual impact of keratinocytes on neural transmission of cold stimuli can be debated.

We agree that in our original title was an overstatement in the absence of any sensory neuron electrophysiology. We have updated our title to that suggested by the reviewers “Keratinocytes contribute to normal cold and heat sensation”.